# Variant Arterial Supply of the Descending Colon by the Coeliac Trunk: A Case Report

**DOI:** 10.3390/medicina57050487

**Published:** 2021-05-12

**Authors:** Sandra Petzold, Silke Diana Storsberg, Karin Fischer, Sven Schumann

**Affiliations:** 1Institute of Anatomy, Medical Faculty, Otto-von-Guericke-University Magdeburg, 39120 Magdeburg, Germany; sandra.petzold@med.ovgu.de (S.P.); karin.fischer@med.ovgu.de (K.F.); 2Institute for Anatomy and Clinical Morphology, School of Medicine, Faculty of Health, Witten/Herdecke University, 58448 Witten, Germany; silke.storsberg@uni-wh.de; 3University Medical Center, Institute for Microscopic Anatomy and Neurobiology, Johannes Gutenberg-University, 55131 Mainz, Germany

**Keywords:** left colic artery, aberrant left colic artery, common hepatic artery, arterial variations, mesenteric arteries, large intestines

## Abstract

*Background and Objectives*: Knowledge of arterial variations of the intestines is of great importance in visceral surgery and interventional radiology. *Materials and Methods*: An unusual variation in the blood supply of the descending colon was observed in a Caucasian female body donor. *Results*: In this case, the left colic artery that regularly derives from the inferior mesenteric artery supplying the descending colon was instead a branch of the common hepatic artery. *Conclusions*: Here, we describe the very rare case of an aberrant left colic artery arising from the common hepatic artery in a dissection study.

## 1. Introduction

Accurate knowledge of large intestine vascular anatomy is of fundamental importance, particularly in visceral surgery and interventional radiology. Variations in vascular anatomy are an important risk factor for severe bleedings during surgery and postoperative complications such as ischemia or anastomotic insufficiency. Additionally, variations in the vascular supply of organs can influence the symptoms following vascular occlusion. In classic anatomical and surgical textbook descriptions, the large intestine is supplied by the superior and inferior mesenteric arteries (SMA and IMA) [1,2,3,4]. The SMA regularly gives rise to the ileocolic (ICA), right colic (RCA), and middle colic (MCA) arteries, which supply the caecum, vermiform appendix, ascending colon, and proximal two-thirds of the transverse colon. The IMA supplies the distal third of the transverse colon, descending and sigmoid colon, the upper two thirds of the rectum, and the anal canal via the left colic, sigmoid, and superior rectal arterial branches, respectively. The left colic artery (LCA) usually arises from the IMA shortly after its origin. After a short descending course, the LCA divides into an ascending and a descending branch. The ascending branch travels to the splenic flexure and anastomoses with the left branch of the MCA via the marginal artery of Drummond within the transverse mesocolon. The descending branch of the LCA passes downwards and anastomoses with the uppermost sigmoid artery [2]. Occasionally, the LCA gives rise to a branch shortly after its origin, which ascends in the radix mesenterii and anastomoses directly with the MCA (Riolan´s arc or arcus magnus mesentericus of Haller [5]).

## 2. Materials and Methods

An unusual variation in the blood supply of the descending colon was observed in an 81-year-old Caucasian female who died of a heart attack. We lack any further clinical information. The woman was part of the body donation program of the Institute of Anatomy, Otto-von-Guericke-University Magdeburg, Magdeburg, Germany and donated her body voluntarily for medical education and research. The donor was fixed via arterial perfusion with ethanol and formaldehyde and subsequent immersion within an ethanol bath. Dissection was performed during the routine dissection classes for undergraduate medical students at the Institute of Anatomy, Otto-von-Guericke-University Magdeburg, Magdeburg, Germany.

## 3. Case Report

In the presented case, a regular LCA that regularly derives from the IMA was instead a branch of the common hepatic artery (CHA). The coeliac trunk (CT) was not formed as a Tripus Halleri but as a Dipus with a left gastric artery (LGA), leaving the CT before bifurcation into the LCA and splenic artery (SA). The variant, i.e., aberrant LCA (ALCA) coursed beneath the pancreas and portal vein. A branch to the pancreas rose from the ALCA. A distinct dorsal pancreatic artery (DPA) from the SA, the CT, or the CHA was not found. The ALCA divided into one ascending branch to the splenic flexure and one descending branch to the descending colon. These branches anastomosed with the sigmoid arteries (SiA) from the IMA and with the MCA from the SMA forming a marginal artery (Figure 1 and Figure 2). The distribution pattern of the SMA was regular. No obvious pathologies or other anatomical variations of the gastrointestinal tract were found. No signs of surgical intervention were noted on the gastrointestinal tract.

## 4. Discussion

Variations of the large intestines’ vascular system are common in general, but are described with heterogeneous prevalence due to the different techniques (e.g., anatomical dissection, angiography) and analyzed populations [6]. According to standard textbooks, the LCA is a branch of the IMA [1,2,3,4]. This standard pattern of origin is present in about 89% of cases. In 25% of cases, the IMA trifurcates into the LCA, sigmoid arteries (SiA), and superior rectal arteries (SRA). In about 39% of cases, the LCA arises from a common trunk with the SiA [6]. Absence of the LCA is rarely found in humans, with frequencies ranging from 0.7% to 12% [7]. In a cohort of 156 individuals, Vandamme reported a remarkable sex dimorphism concerning the absence of the LCA. Whilst the LCA was absent in 16% of all males investigated, this artery was absent only in 6% of the analyzed female bodies [8]. In a meta-analysis of 19 studies (*n* = 2040 patients) the prevalence of LCA absence was only 1.2% [9]. When the LCA is absent, the descending colon and the splenic flexure can be supplied by Turner´s plexus and/or other vessels such as the MCA or the sigmoid arteries [10,11]. In rare cases, the LCA can derive from vessels other than the IMA. In these cases, the LCA should be called an aberrant left colic artery (ALCA). For example, the LCA is a branch of the SMA in less than 1% of cases, and directly derives from the abdominal aorta in less than 0.1% [6].

An accessory left colic artery (AcLCA) is an additional artery supplying the left site colon. In general, the AcLCA is supposed to be a rare condition. Nevertheless, some authors reference remarkably high incidences of up to 49.2%, possibly due to confusion with an accessory middle colic artery (AcMCA) or SiA [12]. With an AcLCA, the proper LCA can be absent, atrophic, or displaced [8]. The AcLCA can derive directly from the IMA or from one of its branches, but there are also AcLCAs rising from sources other than the IMA [13]. According to Kachlik and Hoch, an accessory LCA from an unusual origin should be called accessory aberrant left colic artery (AcALCA) [14].

Michels described the rare case of an accessory aberrant middle colic artery (AcAMCA) from the dorsal pancreatic artery (DPA), which derived from the CHA and anastomosed with the LCA [15]. The DPA regularly derives from the SA (in 40% of cases), but it can also derive from the CHA (17% of cases) or directly from the CT (28% of cases) [6]. The DPA has a branch to the inferior pancreatic artery at the lower border of the pancreas and anastomoses with other pancreatic arteries (e.g., superior pancreaticoduodenal) [2]. In our present case, the ALCA had a branch to the pancreas. It therefore may have taken over the function of the DPA, since a DPA was missing.

An arterial supply of the splenic flexure from the CT or its branches is described in the literature. A coeliacocolic trunk is a single aberrant MCA (AMCA) that derives from the CT and supplies the transverse colon and the splenic flexure [16]. An AMCA originating from the CHA was described by Wadhwa [17]. This AMCA anastomosed with a regular LCA. A branch from the CHA to the splenic flexure is mentioned in a clinical case report by Wu. Unfortunately, no further information about the LCA is given in this article [18]. An AcALCA from the CT is briefly mentioned by Vandamme, without any further description [8].

Anastomotic vessels between the CT and the LCA are a rare phenomenon [19,20]. These special conditions can be explained by an atypical persistence of parts of the longitudinal ventral anastomosis of Tandler [21]. According to Tandler, the longitudinal ventral anastomosis is a temporary connection between the vitelline arteries, which supply the developing gut tube. It allows for the reorganization of the vitelline arteries to the definitive abdominal arteries. Both the DPA and the LCA derive from the longitudinal ventral anastomosis. Although this might explain the connection of these vessels in the present case [22], the existence of the longitudinal ventral anastomosis is discussed controversially [23].

In our case, the ALCA divided into one ascending and one descending branch. According to Pikkieff, this is the most common branching pattern of an LCA (50%) [24].

Accurate knowledge and intra-operative assessment of the vascular anatomy and possible variations thereof are of paramount importance when performing abdominal surgical interventions. An undetected anomalous course of the LCA beneath the pancreas could potentially cause its injury during pancreas resection, resulting in severe bleeding or colon ischemia (in the case of insufficient perfusion via the marginal artery). Variant branches of the CHA might also complicate hepatic surgery (e.g., liver transplantation) and should be considered in colon surgery (e.g., left hemicolectomy) as well.

## 5. Conclusions

The presented case represents an exceptional case of an aberrant left colic artery (ALCA) from the common hepatic artery (CHA) of the coeliac trunk (CT), replacing the regular left colic artery (LCA) originating from the inferior mesenteric artery (IMA).

## Figures and Tables

**Figure 1 medicina-57-00487-f001:**
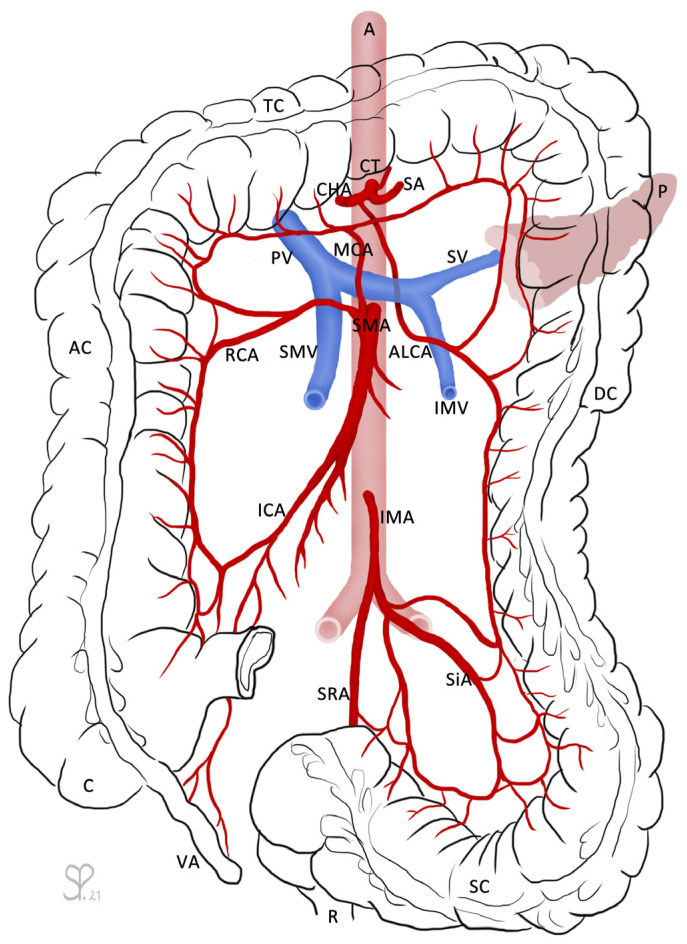
The large intestines (caecum (C) with vermiform appendix (VA), ascending colon (AC), transverse colon (TC), descending colon (DC), sigmoid colon (SC)), as well as the rectum (R) and pancreas (P) are outlined for orientation. From the abdominal aorta (A) arises the coeliac trunk (CT), which branches into the common hepatic artery (CHA), the splenic artery (SA), and the left gastric artery (not labeled). The aberrant left colic artery (ALCA) descends shortly after the branching from the CHA and runs underneath the pancreas and splenic vein (SV). The SV receives the inferior mesenteric vein (IMV) and fuses with the superior mesenteric vein (SMV) to become the portal vein (PV). The superior mesenteric artery (SMA) branches into the middle colic artery (MCA), right colic artery (RCA), and ileocolic artery (ICA). The inferior mesenteric artery (IMA) divides into the sigmoid arteries (SiA) and superior rectal artery (SRA). The ALCA forms anastomoses with the MCA via its ascending branch and with the SiA via its descending branch. Drawing by Sandra Petzold.

**Figure 2 medicina-57-00487-f002:**
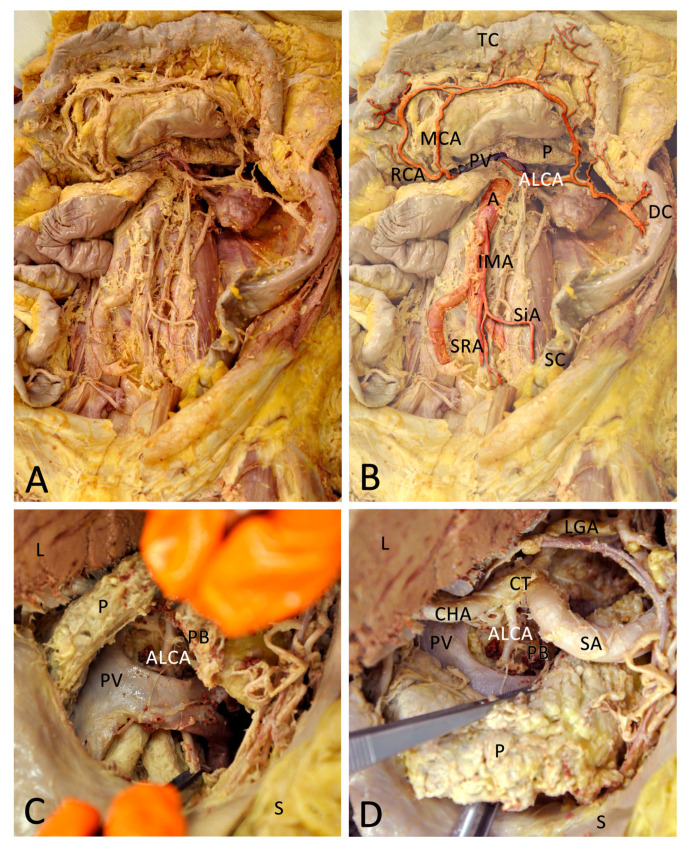
(**A**) Overview of the dissected area. (**B**) The arteries and portal vein (PV) are coloured and contrasted to highlight important areas, while the background is faded. The aberrant left colic artery (ALCA) appears underneath the PV and pancreas (P) to supply the descending colon (DC). The ALCA forms anastomoses with the middle colic artery (MCA) via its ascending branch and with the sigmoid arteries (SiA) via its descending branch. The right colic artery (RCA), as well as the MCA, derives from the superior mesenteric artery (not pictured). The inferior mesenteric artery (IMA) derives from the abdominal aorta (**A**). The IMA divides into the SiA, which supply the sigmoid colon (SC) and the superior rectal artery (SRA). (**C**) View through the dissected lesser omentum (omentum minus). Below the stomach (S) is visible and up is the cut liver (L), the pancreas is pulled up. The descending course of the ALCA (in forceps) beneath the splenic vein that fuses with the superior mesenteric vein to the PV is clearly visible. The ALCA supplies a pancreatic branch (PB). (**D**) Now the pancreas is pulled down to show the coeliac trunk (CT) and the origin of the ALCA. The lower forceps are still fixating the ALCA. The CT divides into the left gastric artery (LGA), splenic artery (SA), and common hepatic artery (CHA). Shortly after the branching, the ALCA derives from the CHA.

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
