# Peer review of "Variant Arterial Supply of the Descending Colon by the Coeliac Trunk: A Case Report"

_medicina, 2021, doi:10.3390/medicina57050487_

Round 1

Reviewer 1 Report

Please see edits and comments included with edited manuscript. I do not agree that the vessel highlighted in this article should be an accessory aberrant left colic artery, but rather an aberrant left colic artery. The term 'accessory' is usually used in describing an additional vessel being present when the referenced vessel (in this case the left colic artery) is also present. The vessel being described as 'accessory' is an additional vessel supplying the same area as the referenced vessel. You define the accessory aberrant left colic artery in this manner in the manuscript, but then use it incorrectly to describe the aberrant vessel being discussed. For this reason, I have listed that this is a major revision that needs editing since it involves the key vessel being discussed in the manuscript.

Reviewer 2 Report

This is a very interesting case report and above all I congratulate the authors for their excelent dissection, photodocumentation and figures. The main weakness of the report is the discussion, which needs to be developed. I suggest adding clinical correlations,  embryology and a more comphrehensive disscussion about the left colic artery. 

One important point that the authors should stress in their report and be humble about is that a left colic artery arising from the coliac trunk is a known variation (although rare it has been documented) and that a case of a LCA arising proximally from the CHA is almost no different from an origin from the coeliac trunk (arteries same course, origins are only a few centimeters away from each other). 

  1. Clinical correlations
    The authors should mention which in particular clinical scenarios might this vessel have significance. For example this vessel is certainly liable to being ligated during pancreatic resections, as it is in essence replacing the dorsal pancreatic artery (I presume a dorsal pancreatic artery was not found?). However, if good collateral circulation exists via the marginal artery of Drummond, this is unlikely to lead to serious consequences. This should be discussed.
  2. Embryology
    For cases like this at least a brief description of possible embryological origin of the variation. E.g. longitudinal anastomosis and Tandler's theory. I would recommend reading this article - 
    https://link.springer.com/article/10.1007%2Fs00276-006-0098-8

  3. Left colic artery

    A good overview is provided the monograph by Kachlik and Hoch, which the authors already have referenced. The left colic artery can arise from the inferior mesenteric artery either alone or with other arteries as a common trunk, it can arise also from the superior mesenteric artery or coeliac trunk. Michel's monograph should be referenced and the variation should be compared to his coeliacocolic trunk (he had one case of a LCA from a coeliac trunk), which as I mentioned above is the most similar previously reported variation to the current cases.

    There are also various difference courses and branching patterns the left colic artery can take and these should also be mentioned briefly (again see Kachlik and Hoch, also Michels and Pikkieff).  The course and branching of the current case should also be described. 

    There have also been cases of LCA arising from a dorsal pancreatic artery - but I would rebut these cases - and rather would favour Michels explanation who considere cases of colic arteries arising from the coeliac trunk as enlarged dorsal pancreatic arteries  - see Michels monograph on the blood supply of the upper abdominal organs - particularly, his description of the coeliacocolic trunk. 

As a last note I would refain from using phrases such as " this is the only reported case" as such phrases cannot be proven. 

I have also made some comments and language corrections directly to the pdf, which the assistant editor will forward to you. I hope you will find my suggestions helpful and that they will improve the article. Of course I do not insist that you have to follow my every suggestion, but I will be happy if the discussion is rewritten, more developed and better structured. 

Round 2

Reviewer 1 Report

There are a few minor grammatical errors that I have addressed below since I am unable to edit the PDF or converted Word document. Other than that, the edits made by the authors are excellent and with these few errors fixed, the manuscript is approved on my end.

Grammatical errors:

1) Line 28: Add a comma after descriptions to say "descriptions, the"

2) Line 30: Add a comma after arteries to say "arteries, which"

3) Line 38: Delete extra space between "branch" and "of"

4) Line 135: Delete extra space between "and" and "pancreas"

5) Line 138: Delete extra space between "artery" and "IMA" as well as "from" and "the"

6) Line 184: I do not see that AcALCA is defined in the document prior to its mention here, so please define it in parentheses before AcALCA is mentioned.

Author Response

Dear reviewer,
we included the comments and linguistic remarks in our manuscript. We highly appreciate the support of the editors in the revision process. Best regards.